# A Study of Contrastive Learning Algorithms for Sentence Representation Based on Simple Data Augmentation

Xiaodong Liu, Wenyin Gong ⬤, Yuxin Li *, Yanchi Li ⬤ and Xiang Li ⬤

School of Computer Science, China University of Geosciences (Wuhan), Wuhan 430079, China; liuxiaodong@cug.edu.cn (X.L.)
* Correspondence: isliyuxin@cug.edu.cn; Tel.: +86-13340390396

**Abstract:** In the era of deep learning, representational text-matching algorithms based on BERT and its variant models have become mainstream and are limited by the sentence vectors generated by the BERT model, and the SimCSE algorithm proposed in 2021 has improved the sentence vector quality to a certain extent. In this paper, to address the problem that the SimCSE algorithm has—that the greater the difference in sentence length, the smaller the probability that the sentence pairs are similar—an EdaCSE algorithm is proposed to perturb the sentence length using a simple data enhancement method without affecting the semantics of the sentences. The perturbation is applied to the sentence length by adding meaningless English punctuation marks to the original sentence so that the model no longer tends to recognise sentences of similar length as similar sentences. Based on the BERT series of models, experiments were conducted on five different datasets, and the experiments proved that the EdaCSE method improves an average of 1.67, 0.84, and 1.08 on the five datasets.

**Keywords:** BERT; data augmentation; sentence vector representation

## 1. Introduction

Currently, the text-matching technique based on sentence representation is one of the mainstream text-matching algorithms, which converts two sentences into a vector representation and measures the degree of similarity of the sentences by calculating the similarity between the two vectors. A good sentence vector representation should satisfy two conditions: one is alignment, which means that similar sentence vectors should have similar distances in the vector space. The second is homogeneity, which means that word vectors should be uniformly distributed in the vector space. The word vectors produced by native BERT [1] are anisotropic and affected by word frequency. The word vector space is vertebrate-shaped, with high-frequency words clustered in a small region in the space (the top of the cone) and low-frequency words dispersed at the bottom of the vertebra. The spatial distribution is not only uneven, but a large part of the space is vacant and does not represent any semantics, so the sentence vectors produced at this time are of low quality and have no way to characterise the sentence semantics well. The SimCSE [2] (Simple Contrastive Learning of Sentence Embeddings) algorithm, proposed in 2021, improves the quality of sentence vectors to some extent. extent improves the quality of sentence vectors.

In this paper, we will study the sentence representation-based text-matching algorithm based on the BERT family of models, analyse the advantages and disadvantages of SimCSE, and aim to construct a better sentence vector representation algorithm.

## 2. Related Work

Traditional sentence vector standard-type algorithms perform post-processing on top of BERT-generated sentence vectors without directly optimising for the BERT model itself, and these methods usually neglect how to make the BERT model itself generate better sentence vectors. However, due to the feature space collapse problem in BERT models,

researchers have started to focus on contrast learning methods to solve the feature space collapse problem [3]. Contrastive learning aims to distance the vectors between dissimilar samples while bringing the vectors between similar samples closer together to learn a higher-quality representation [4]. The method forces the model to cluster similar samples in the embedding space by introducing a similarity metric into the training process in order to better learn a more discriminative feature representation.

Methods that use contrast learning to solve the feature space collapse problem have made significant progress in improving the quality of sentence representations compared with traditional standard-type algorithms for sentence vectors. These approaches have enabled models to focus more on similarity and differentiation when generating sentence vectors, thus enabling better application to a variety of natural language processing tasks.

Yan Y et al. [3] proposed a contrast learning framework, ConSERT, using BERT as a shared encoder with the last layer output averaged and pooled as a sentence vector, and proposed data enhancement strategies such as Dropout, constructing adversarial samples based on gradients, randomly cropping words or features, and disrupting word order to construct positive and negative samples for contrast learning. Experimentally, ConSERT proved to be effective in ameliorating the problem of collapsed representations of native BERT sentences. Gao T et al. [2] proposed a simple contrast learning framework, SimCSE, with two training approaches: an unsupervised approach, where two different representations of the same sentence are obtained as positive sample pairs through the Dropout mechanism and constitute negative sample pairs with other sentences in the same batch; and a supervised approach, where using the NLI dataset, sentence pairs with embedded relations are used as positive sample pairs and sentence pairs with contradictory relations are used as negative sample pairs for comparison training.

The ESimCSE proposed by Wu et al. in 2022 [5] addresses two problems: (1) positive examples constructed using Dropout have the same sentence length, leading the model to consider sentences of the same length as more similar; and (2) increasing the batch size introduces more negative examples, which in turn leads to a decrease in effectiveness. To address these two issues, ESimCSE uses repeated words to construct positive examples and momentum sequences to expand negative samples. The ArcCSE proposed by Zhang Y et al. [6] addresses two problems: (1) vulnerability to noisy data interference and (2) inability to model the semantic order between multiple sentences. To address these issues, ArcCSE converts the NT-Xent objective function, which previously operated in Euclidean space, to angle space and adds angle decision boundaries, resulting in a more robust sentence representation. In addition, to model the semantic order between multiple sentences, ArcCSE proposes a new pre-training task to model the implication relations of ternary sentence pairs. Jiang T et al. [7] proposed Sentence-T5 (abbreviated as ST5) to train a sentence vector model by contrast learning using CommQA and NLI datasets with a twin-tower structure. Experiments show that the average pooling result of the ST5 encoder output works best as a sentence vector. Klein T et al. [8] proposed a joint self-contrast learning and feature de-correlation method, SCD. On the semantic similarity task, SCD does not have an advantage over SimCSE but has a respectable performance on several other downstream tasks.

Chuang Y S et al. [9] proposed DiffCSE, an unsupervised contrast learning framework, whose research idea comes from the CV domain, i.e., for input samples, contrast loss can be constructed based on insensitive transformations and prediction loss based on sensitive transformations, and this approach helps to improve the representational power of the model. SNCSE was proposed by Wang H et al. [10], aiming to solve the problem of feature suppression caused by data enhancement methods; SNCSE introduces soft negative examples and bi-directional marginal loss to limit the range of similarity. Meanwhile, SNCSE acquires word vectors through cue learning and syntactic parsing using spacy and constructs soft negative examples. ease was proposed by Nishikawa S et al. [11] to enhance the learning of sentence vectors by using entity information. Ease constructs positive samples by hyperlinking entities from Wikipedia, and negative entities need to have the

same type as positive entities and cannot be on the same Wikipedia page. Randomly selected candidate entities that satisfy the conditions are used as hard negative example data to construct the ternary data.

Scholars have researched sentence vector representation algorithms based on contrast learning from different directions and have achieved rich research results. In this paper, based on the SimCSE algorithm, we will analyse its current shortcomings and make improvements.

### 3. Contrastive Learning Algorithm for Sentence Representation Based on Simple Data Augmentation

*3.1. SimCSE Introduction*

3.1.1. Comparative Learning

Contrastive Learning is a self-supervised learning method based on learning a compact representation of data. The basic idea of Contrastive Learning is to bring a pair of similar data as close together as possible by comparing them while pushing them as far away as possible from other data. This allows similar data to be brought closer together in the representation space and dissimilar data to be moved away. Contrast learning is commonly used for feature learning on unlabeled data, especially in the fields of computer vision and natural language processing.

The core of contrast learning is the loss function, which is usually employed, such as InfoNCE loss [12] or triplet loss [13]. InfoNCE loss measures the difference between positive and negative samples in the feature space by calculating their relative probabilities. An effective feature representation is learned by adjusting the feature representation so that the relative difference between positive and negative samples is minimised. Triplet loss is constructed in the form of a triad (samples, positive instances, and negative instances) so that the positive samples are as close as possible and the negative samples as far away as possible in the feature space.

InfoNCE Loss was adopted by SimCSE into the sentence representation, as shown in Equation (1):

$$\text{Contrastive Loss}_i = -\log\left(e^{S(z_i, z_j^+)/\tau} \middle/ \sum_{j=0}^{K} e^{S(z_i, z_j)/\tau}\right) \tag{1}$$

where K represents the amount of data in a batch, $z_i$ represents the total samples i in this batch, Contrastive Loss$_i$ represents the loss value of sample i, $S\left(z_i, z_j^+\right)$ represents the cosine similarity between the sample and its similar samples, and $\tau$ represents the temperature parameter, which is used to control the degree of attention to the difficult negative samples, and the larger the value is, the greater the attention to the difficult negative samples. $\sum_{j=0}^{K} e^{S(z_i, z_j)/\tau}$ represents the sum of similarity values between sample i in a batch and samples within other batches. Overall, the goal of the loss function is to make the numerator value larger and the denominator value smaller, z, that is, to make the distance between positive samples increase and the distance between negative samples decrease.

3.1.2. SimCSE Theory

SimCSE (Similarity-based Contrastive Self-supervised Learning) is a contrastive learning-based framework for learning sentence representations. SimCSE is not a new model; it proposes two contrastive learning-based agent tasks to learn sentence representations in unsupervised and supervised ways.

SimCSE's first agent task was inspired by Dropout, an anti-overfitting technique in neural networks where Dropout randomly closes the connections of some neurons, which results in different closed neurons and different outputs from the final model. Based on this observation, SimCSE obtains two sentence embeddings by passing the same

sentence through the same model twice but using different Dropout operations. These two embeddings are treated as positive examples of the model, while other embeddings in the same batch are treated as negative examples.

The second agent task of SimCSE directly employs a supervised dataset of NLI (Natural Language Inference) for comparative learning training. The goal of the NLI task is to determine the relationship between two sentences, and the possible relationships include entailment, contradiction, or neutral. In this task, sentence pairs with entailment can be directly used as positive examples. Meanwhile, other embeddings in the same batch can be used as negative examples. By comparing the similarities and differences between positive and negative examples, the model can learn to encode semantically similar sentence pairs as similar embeddings. In addition, the authors try to add hard negative examples to the negative examples, here referring to sentence pairs of contradiction in the NLI dataset. By introducing contradictory sentence pairs as negative examples, the difficulty of contrast learning can be further increased, thus improving the performance of the model.

SimCSE solved the problem of characterising anisotropy in native BERT word vectors to a certain extent while achieving a substantial lead on the major sentence similarity tasks, making it the state-of-the-art model of its time.

### 3.2. Existing Problems

In SimCSE, positive samples are constructed in such a way that sentence vectors obtained from the same sample encoded twice by the same encoder are positive samples of each other. Therefore, under this model of contrast learning, all positive samples are constructed as samples of the same length, which may introduce a bias to the model by incorrectly assuming that the more similar the lengths of the samples are, the higher the probability of being a positive sample pair, and the higher the difference in lengths, the lower the probability of sentence pairs being similar. Such a bias may negatively affect the performance of the model in some scenarios.

Three sets of cases are shown in Table 1. The sentences in the column Sentence1 are taken from the training set. The sentences in the column Sentence2 are processed by adding irrelevant words to the sentences in Sentence1. Row1, Row2, Row3, and Row4 show negative sample pairs, and Row5 and Row6 show positive sample pairs. It can be seen that for both positive and negative sample pairs, as long as the length of the two sentences is convergent, the similarity is improved. It can be seen that the words added here are completely meaningless and should not interfere with the relationship between the sentences, so the model will "unintentionally" learn this kind of misinformation during the training process.

**Table 1.** Offset display.

| Row | Sentence1 | Sentence2 | Cosine Similarity |
| --- | --- | --- | --- |
| 1 | How to type the currency symbols on computer? | What are the currency symbols? | 0.388 |
| 2 | How to type the currency symbols on computer? | What are the currency symbols? Okay, okay, okay. | 0.437 |
| 3 | Billy Billy Bateson appeared in the first four issues of Black Adam, published from late 2008 to early 2009. | He moved back to Philadelphia in 2009 and now lives in New York City. | 0.102 |
| 4 | Billy Billy Bateson appeared in the first four issues of Black Adam, published from late 2008 to early 2009. | He moved back to Philadelphia in 2009 and now lives in New York City. What are the currency symbols for each country? What | 0.154 |
| 5 | Can I cancel if I borrowed money but it hasn't gone through yet? | Can it be cancelled? | 0.8784 |
| 6 | Can I cancel if I borrowed money but it hasn't gone through yet? | Can it be cancelled? ? ? ? ? ? | 0.8845 |

*3.3. Algorithm Design*

In this paper, we propose a simple data augmentation-based Contrastive Learning of Sentence Embeddings (EdaCSE), an algorithm for the comparative learning of sentence representations. The core idea of the method is to perturb the length of positive sample pairs without making changes to the meaning of the sentences.

Nowadays, BERT and its variants are prevalent in major tasks; however, conventional data enhancement methods for BERT, such as replacing synonyms, disrupting the order of sentences, and back-translation methods, are sensitive transformations, and these data enhancement methods cause the semantics of the sentence vector representations generated by the model to be highly biassed, impairing the performance on downstream tasks. The emergence of Adversarial Learning [14] addresses such problems to a large extent. In adversarial learning, generally labelled data is required, and the basic process is to have the model forward propagate once, then calculate the loss for backpropagation, then calculate a perturbation value, add this perturbation value to the parameters, and then perform forward propagation and back propagation again, and the gradients obtained from the two times are summed up, restoring the model to what it was before the addition of the perturbation and updating it. This method serves as data augmentation and also enhances the robustness of the BERT model, but there are problems: (1) labelled data is required; (2) backpropagation has to be performed twice for the same sample; (3) and the training time is greatly increased.

Therefore, this paper explores the non-sensitive enhancement of sentences without labelling. It was found that randomly adding English punctuation marks "." in Chinese sentences, ",", "!", "?", ";" and ":" can achieve the above goal. As shown in Table 2, before and after the application of data augmentation to a sentence, it does not literally affect the meaning of the original sentence to a large extent. In Section 5.3, the strongly intonated symbols "?" and "!" were further explored to see if they interfered with the meaning of the sentences.

**Table 2.** Sentences before and after applying data augmentation methods.

| Original Sentence | Sentences after Applying Data Augmentation |
|---|---|
| Why can't I open Microparticle Loan on my mobile phone | Why. can't I open: Microparticle Loan on my mobile phone |
| Failure to meet borrowing requirements | Failure to meet borrowing requirements; |
| This high-resolution picture, who has it | This, high-resolution picture, who has it? |
| How's Deng's singing | ? How's Deng's: singing |

Figure 1 shows the structure of the EdaCSE algorithm.

As can be seen from Figure 1, the model receives two inputs: one is the original sample, and the other is the sample after simple data augmentation. The output generated by the original sample is used to calculate the contrast loss according to the model in SimCSE, while the output generated by the sample after data augmentation is used as a candidate sample, and the sentence vector representation of the original sample and its sentence vector representation after data augmentation are mutually positive samples. The sentence vector representations produced after data augmentation with other data in the batch then form a negative sample, and in this way, an additional augmented contrast loss is calculated, which is the Contrastive Loss$_{\text{EdaCSE}}$ in Equation (2), and the two losses are then combined in some proportion to form the EdaCSE algorithm.

$$\text{Loss} = \text{Contrastive Loss}_{\text{CSE}} + \lambda * \text{Contrastive Loss}_{\text{EdaCSE}} \tag{2}$$

Here, the Contrastive Loss is calculated as shown in Equation (1), where Contrastive Loss$_{\text{CSE}}$ represents the loss calculated by using the original SimCSE and Contrastive Loss$_{\text{EdaCSE}}$ represents the loss calculated by using the EdaCSE algorithm. Both of them are

calculated in the same way, but the latter calculates the loss by constructing comparative learning between the simple data-augmented samples and the original samples as positive example pairs.

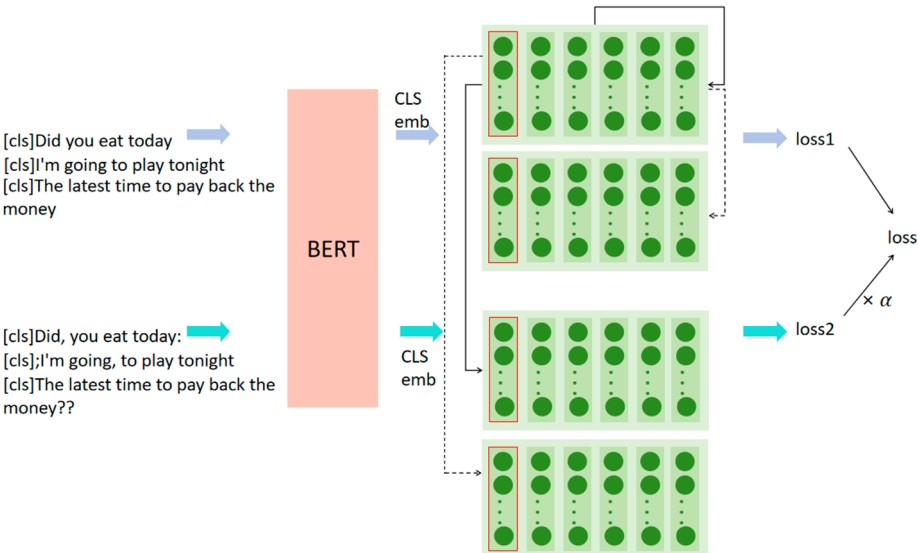

**Figure 1.** Illustration of the EdaCSE algorithm.

The samples augmented with simple data are used as positive examples to construct comparative learning with the original sentences, and similarly, additional information to the model is introduced to correct for bias in SimCSE and enhance generalization. Contrastive Loss$_{EdaCSE}$ is the loss value generated by the EdaCSE method, and λ is a hyperparameter with values from 0 to 1.

## 4. Description of the Experiment

### 4.1. Experimental Environment Setting

The hardware and software environments applied for the experiments in this paper are shown in Table 3 below.

**Table 3.** Experimental hardware and software configuration.

| System Development Environment | Development Environment | CUDA Version | GPU | CPU | Memory Size |
|---|---|---|---|---|---|
| Ubuntu20.04 | Pycharm2020.1 | 11.3 | NVIDIA RTX A5000 (24 G) | AMD EPYC 7551P | 64 G |

The experimental Chinese text splitting and pre-training models use Transformers, an open-source NLP model library provided by the Huggingface community, which provides a range of pre-training models and corresponding tools that can be used in a variety of deep learning frameworks. In addition, the experiments used Python 3.8 as the programming language, Pytorch as the deep learning framework, and adamW as the optimizer [15]. All experimental results were evaluated using the Spearman correlation coefficient as a metric.

### 4.2. Evaluation Metrics Setting

Spearman's correlation coefficient is a commonly used similarity measure algorithm for evaluating the relevance of a non-linear relationship between two variables. Compared with other similarity measures such as cosine similarity, Minkowski distance, and Pearson's correlation coefficient, Spearman's correlation coefficient has the following characteristics:

(1) Non-parametric approach: the Spearman correlation coefficient is a non-parametric statistic that does not depend on the specific form of the distribution of the data, which

makes it effective in assessing correlation on a wide range of data types (continuous, discrete, ordered, etc.).

(2) Suitable for ordered data: the Spearman correlation coefficient is particularly suitable for ordered or ranked data, and it is able to alleviate the limitations of other measurement algorithms on this type of data by comparing the rank order of the variables.

(3) Evaluation of non-linear relationships: Unlike the Pearson correlation coefficient, which applies only to linear relationships, the Spearman correlation coefficient is able to capture the correlation of non-linear relationships, which makes it more flexible in practical applications and allows for a more comprehensive description of the interrelationships between variables.

Due to the applicability and flexibility of Spearman's correlation coefficient, Spearman's correlation coefficient was chosen as the evaluation metric for the subsequent experiments in this paper.

The Spearman's correlation coefficient is calculated as shown in Equation (3), which takes values between $-1$ and 1, where $-1$ indicates a perfect negative correlation, 1 indicates a perfect positive correlation, and 0 indicates no correlation. Positive values indicate that as one variable increases, the value of the other increases; negative values indicate that as one variable increases, the value of the other decreases. The Spearman correlation coefficient can be used to measure the degree of similarity between two sentence vectors.

$$\rho = 1 - \frac{6 \sum d_i^2}{n(n^2 - 1)} \tag{3}$$

where $d_i$ denotes the difference in ranks between the ith elements of X and Y, and n denotes the length (i.e., the number of elements) of X and Y.

*4.3. Datasets*

4.3.1. Basic Introduction to Datasets

The experiments used five datasets: ATEC, BQ, LCQMC, PAWS-X, and Chinese STS-B, which are briefly described here.

ATEC-Question Similarity Calculation is a competition question hosted by Ant Financial Services that uses an algorithm to determine whether two given sentences have the same semantic meaning, which can be applied in a customer service system, and all its data comes from actual application scenarios of Ant Financial Brain. The dataset has 62,477 training samples, 20,000 samples in the development set, and 20,000 samples in the public test set.

The Bank Question (BQ) dataset [16] is a large-scale domain-specific Chinese dataset for Sentence Semantic Equivalence Recognition (SSEI). The BQ dataset contains 120,000 question pairs from the online banking customization service logs. It is divided into three parts: 100,000 pairs for training, 10,000 pairs for validation, and 10,000 pairs for testing.

LCQMC [17] is a large-scale Chinese question-matching corpus covering a large number of question-pair data. The corpus contains 260,068 question pairs with manual annotations and is divided into a training set, a development set, and a test set. The training set contains a total of about 238,766 question pairs, the development set contains a total of about 8802 questions, and the test set contains a total of about 12,500 questions.

PAWS-X [18] is a new dataset consisting of 23,659 human-translated PAWS evaluation pairs in six different types of languages: French, Spanish, German, Chinese, Japanese, and Korean. PAWS-X demonstrates the effectiveness of deep, multilingual pre-training while leaving considerable scope to drive multilingual research that better captures structural and contextual information. The number of training set samples in this dataset is 49,129, with a positive and negative sample count of 27,445 and 21,684, respectively.

STS-B [19] (Semantic Textual Similarity Benchmark) is a dataset for measuring textual similarity. sts-b contains a set of sentence pairs about textual similarity, each with a score that measures how similar the two sentences are semantically. The score is a real number from 0 to 5, where 0 means that the two sentences are not related at all and 5 means that the two sentences are identical or very similar. The Chinese STS-B dataset is mainly translated from the original English STS-B dataset and is used for fine-grained Chinese text-matching tasks or Chinese natural language inference tasks. The Chinese STS-B contains a total of 5231 training samples, 1458 validation samples, and 1361 test samples.

4.3.2. Training Sample Size Exploration

For the experiments in this paper, the training set parts from the five datasets ATEC, BQ, LCQMC, PAWS-X, and Chinese STS-B were used for fairness in comparison, and only sentences were used as the training corpus; no labels were involved. The data in the five datasets were disrupted and integrated, and the duplicate sentences were removed to obtain a corpus of size 438,650. This corpus will be used as the training set for this paper. The trained model is evaluated by calculating the Spearman correlation coefficient on the open test set in each dataset.

The experiments are trained using the whole corpus as the training set, and the Sim-CSE approach is used as the base method for tuning the reference to "bert-base-chinese", "chinese-roberta-wwm-ext [20]", "chinese-roberta-wwm-ext-large [21]" as the base models. For convenience, the subsequent content in this paper will refer to the three models mentioned above as "bert-base", "roberta-base", and "roberta-large", respectively. The same goes for tables and images. In order not to damage the performance of the model itself, the learning rate range was set to $1 \times 10^{-6}$~$5 \times 10^{-5}$, the batch size range was set to 8~128, and the Dropout range was 0.1~0.8. Hyperparameter search was performed using grid search, and the best hyperparameter settings were obtained based on performance on the test sets of the five datasets, as evaluated in Table 4.

**Table 4.** Training hyperparameter settings.

| Model | Learning Rate | Batch Size | Drop Out Rate | Weight Decay | Learning Rate Decay Strategy |
|---|---|---|---|---|---|
| base | $3 \times 10^{-5}$ | 32 | 0.1 | 0.01 | Cosine decay |
| large | $5 \times 10^{-6}$ | 16 | 0.1 | 0.01 | Cosine decay |

The base in the table corresponds to the "bert-base" and "roberta-base" models, and the large corresponds to the "roberta-large" model.

It was found that the loss value of the model had become zero after a certain number of samples were trained in one epoch, so we continued to explore the appropriate number of training samples. Due to the large randomness of sample extraction, the samples in each dataset may not be drawn evenly, so during the experiments, the best performance was taken as the result of each group of experiments after ten trials, and this criterion was followed in subsequent experiments.

Table 5 shows the results of the training sample part of the search for bert-base. It can be seen that as the number of training samples decreases from the full data set to 10,000 data points, the performance on the test set generally shows an increase, and below 10,000 it shows a decrease.

Table 6 shows the results of the roberta-base training sample exploration experiments. It can be seen that the model performance of the training sample generally decreases from using the full amount of data to using 10% of the full amount of data, and the model performance is more volatile when the number of samples is taken from 30,000 to 10,000, and the best performance is achieved at 10,000; it still shows volatility when 8000 to 3000, and the model still has good performance when 3000 samples are taken for training.

**Table 5.** Bert-base training sample size exploration.

| Sample Size | ATEC | BQ | LCQMC | PAWS-X | STS-B | AVG |
|---|---|---|---|---|---|---|
| 438,650 | 25.49 | 40.15 | 43.51 | 11.31 | 28.21 | 29.73 |
| 394,785 (90%) | 25.15 | 43.72 | 46.18 | 11.13 | 30.49 | 31.33 |
| 350,920 (80%) | 24.83 | 40.86 | 46.73 | 11.14 | 29.12 | 30.53 |
| 307,055 (70%) | 25.05 | 40.97 | 45.67 | 12.61 | 27.81 | 30.42 |
| 263,190 (60%) | 27.91 | 44.06 | 50.87 | 8.83 | 29.61 | 32.25 |
| 219,325 (50%) | 25.67 | 44.2 | 50.35 | 10.27 | 28.77 | 31.85 |
| 17,544 (40%) | 26.89 | 44.32 | 50.8 | 11.85 | 27.51 | 32.27 |
| 131,595 (30%) | 27.94 | 45.97 | 52.95 | 10.43 | 26.78 | 32.81 |
| 87,730 (20%) | 28.77 | 46.93 | 56.58 | 13.67 | 28.53 | 34.89 |
| 43,865 (10%) | 30.59 | 48.26 | 58.97 | 10.67 | 25.7 | 34.83 |
| 30,000 | 30.33 | 47.85 | 58.95 | 10.92 | 27.41 | 35.09 |
| 25,000 | 31.62 | 48.79 | 58.72 | 9.9 | 26.92 | 35.19 |
| 20,000 | 31.81 | 49.04 | 61.77 | 11.43 | 24.97 | 35.80 |
| 15,000 | 32.09 | 48.25 | 60.37 | 11.31 | 24.51 | 35.30 |
| 10,000 | 32.43 | 48.54 | 63.3 | 10.01 | 25.07 | 35.87 |
| 8000 | 31.65 | 47.88 | 60.03 | 10.48 | 25.3 | 35.06 |
| 5000 | 30.99 | 47.23 | 61.44 | 10.2 | 24.85 | 34.94 |
| 3000 | 29.85 | 45.74 | 64.9 | 10.39 | 24.45 | 35.06 |

**Table 6.** Roberta-base training sample size exploration.

| Sample Size | ATEC | BQ | LCQMC | PAWS-X | STS-B | AVG |
|---|---|---|---|---|---|---|
| 438,650 | 28.52 | 45.57 | 58.21 | 7.3 | 30.37 | 33.99 |
| 394,785 (90%) | 28.65 | 45.65 | 58.73 | 7.53 | 30.55 | 34.22 |
| 350,920 (80%) | 29.31 | 46.29 | 58.06 | 7.61 | 30.33 | 34.32 |
| 307,055 (70%) | 28.42 | 45.98 | 58.17 | 7.38 | 30.08 | 34.00 |
| 263,190 (60%) | 28.54 | 46.08 | 56.95 | 7.82 | 29.91 | 33.86 |
| 219,325 (50%) | 28.26 | 45.54 | 57.81 | 7.81 | 30.79 | 34.04 |
| 17,544 (40%) | 29.38 | 47.06 | 58.55 | 8.46 | 30.62 | 34.81 |
| 131,595 (30%) | 29.93 | 46.57 | 58.36 | 8.99 | 29.8 | 34.73 |
| 87,730 (20%) | 30.71 | 48.25 | 58.39 | 8.28 | 30.23 | 35.17 |
| 43,865 (10%) | 31.16 | 49.99 | 59.38 | 9.97 | 29.39 | 35.97 |
| 30,000 | 32.27 | 49.18 | 64.42 | 9.72 | 29.92 | 37.10 |
| 25,000 | 30.33 | 48.39 | 62.68 | 9.69 | 29.39 | 36.09 |
| 20,000 | 30.53 | 46.84 | 60.43 | 9.53 | 29.37 | 35.34 |
| 15,000 | 32.9 | 49.92 | 66.17 | 10.41 | 29.52 | 37.78 |
| 10,000 | 34.11 | 50.18 | 65.39 | 10.27 | 29.54 | 37.89 |
| 8000 | 32.86 | 50.02 | 64.82 | 10.15 | 29.77 | 37.52 |
| 5000 | 31.99 | 47.23 | 62.44 | 10.2 | 28.85 | 36.142 |
| 3000 | 32.85 | 49.74 | 64.9 | 10.39 | 28.45 | 37.266 |

As can be seen from Table 7, the performance of the roberta-large model varies with sample size in a similar way to the first two models, with the difference that the model performs worse than the first two models if the sample size is less than 10,000. The model still performs best at a training sample size of 10,000.

**Table 7.** Roberta-large training sample size exploration.

| Sample Size | ATEC | BQ | LCQMC | PAWS-X | STS-B | AVG |
|---|---|---|---|---|---|---|
| 438,650 | 30.34 | 46.05 | 63.37 | 8.02 | 30.42 | 35.64 |
| 394,785 (90%) | 30.65 | 46.80 | 64.07 | 8.72 | 30.67 | 36.18 |
| 350,920 (80%) | 30.88 | 47.07 | 64.57 | 8.72 | 30.48 | 36.28 |
| 307,055 (70%) | 30.96 | 46.91 | 64.93 | 8.43 | 30.21 | 36.28 |

**Table 7.** *Cont.*

| Sample Size | ATEC | BQ | LCQMC | PAWS-X | STS-B | AVG |
|---|---|---|---|---|---|---|
| 263,190 (60%) | 30.37 | 46.29 | 63.16 | 8.79 | 30.51 | 35.82 |
| 219,325 (50%) | 30.38 | 47.09 | 63.10 | 9.36 | 30.30 | 36.04 |
| 17,544 (40%) | 30.42 | 47.10 | 64.13 | 9.20 | 30.33 | 36.23 |
| 131,595 (30%) | 31.67 | 47.75 | 65.21 | 9.72 | 30.06 | 36.88 |
| 87,730 (20%) | 31.93 | 48.00 | 65.70 | 9.56 | 30.30 | 37.09 |
| 43,865 (10%) | 32.17 | 48.72 | 67.14 | 10.78 | 30.66 | 37.89 |
| 30,000 | 31.20 | 48.66 | 66.94 | 11.20 | 30.20 | 37.64 |
| 25,000 | 33.19 | 49.47 | 66.95 | 10.83 | 30.16 | 38.12 |
| 20,000 | 31.73 | 48.27 | 67.11 | 11.46 | 29.57 | 37.62 |
| 15,000 | 32.42 | 49.06 | 66.61 | 11.31 | 29.93 | 37.86 |
| 10,000 | 31.42 | 49.09 | 68.15 | 11.67 | 29.75 | 38.00 |
| 8000 | 31.77 | 48.03 | 66.56 | 11.60 | 29.31 | 37.45 |
| 5000 | 22.64 | 39.79 | 66.65 | 11.64 | 29.25 | 33.99 |
| 3000 | 20.89 | 35.33 | 64.88 | 11.61 | 29.82 | 32.50 |

In summary, 10,000 randomly selected samples were chosen as the training set in this paper. The number of training samples for the three models in the other experiments in the later sequence is based on this criterion.

## 5. Experiments and Analysis of Results

### 5.1. Basic Experiment

Table 8 presents the experimental results of the SimCSE algorithm and the EdaCSE algorithm. The models named after the algorithm itself represent the experimental results of combining the model with the SimCSE algorithm, serving as the baseline experiments.

**Table 8.** Comparison of the effects of various methods.

| Model | ATEC | BQ | LCQMC | PAWS-X | STS-B | Mean |
|---|---|---|---|---|---|---|
| bert-base | 32.43 | 48.54 | 63.3 | 10.01 | 25.07 | 35.87 |
| roberta-base | 34.11 | 50.18 | 65.39 | 10.27 | 29.54 | 37.89 |
| roberta-large | 31.42 | 49.04 | 68.15 | 11.67 | 29.75 | 38.00 |
| bert-base + EdaCSE | 34.23 | 50.67 | 64.9 | 10.15 | 27.76 | 37.54 |
| roberta-base + EdaCSE | 33.9 | 51.52 | 67.76 | 10.92 | 29.56 | 38.73 |
| roberta-large + EdaCSE | 33.42 | 51.14 | 69.15 | 11.77 | 29.95 | 39.08 |

From Table 8, it can be observed that using the EdaCSE algorithm, there is an improvement of 1.67 and 0.89 for the "bert-base" and "roberta-base" models, respectively, and 1.08 for the "bert-large" model. This suggests that the use of the EdaCSE algorithm is also effective in improving the generalisation ability of the model. Generally speaking, for models such as BERT, adding and deleting characters is a sensitive transformation that is easy to damage the model performance, and EdaCSE has a favourable impact on the model performance, and in a way, the algorithm improves the robustness of such models. Next, around the EdaCSE method, experimental analyses are carried out on Chinese punctuation, English punctuation, types of punctuation, and the number of punctuation insertions.

### 5.2. Exploring the Number of Inserted Punctuation Marks

First insert 1 to 3 punctuation marks, using the English punctuation marks ".", ",", "!", "?" ",", "!", "?", ";", ":" as a benchmark to explore the number of symbols inserted. Taking the performance of "bert-base" as an example, the effect of the number of symbols on the model performance is shown in Table 9.

**Table 9.** Effect of the number of insertion symbols on the bert-base model.

| Number of Symbols | ATEC | BQ | LCQMC | PAWS-X | STS-B | Mean |
|---|---|---|---|---|---|---|
| 1~1 | 33.38 | 50.51 | 62.00 | 9.46 | 27.29 | 36.52 |
| 1~2 | 32.41 | 51.11 | 64.46 | 11.14 | 26.72 | 37.16 |
| 1~3 | 34.23 | 50.67 | 64.9 | 10.15 | 27.76 | 37.54 |
| 1~4 | 32.68 | 50.17 | 61.98 | 9.27 | 25.98 | 36.01 |
| 1~5 | 33.43 | 50.75 | 63.12 | 10.4 | 24.92 | 36.52 |
| 1~6 | 32.99 | 49.54 | 63.7 | 10.12 | 26.94 | 36.65 |
| 1~7 | 32.95 | 50.1 | 65.44 | 9.78 | 25.8 | 36.8 |

As can be observed from Table 9, the best results are achieved when 1 to 3 punctuation marks are inserted. While the insertion of only one punctuation mark also has a positive effect on the model, the insertion of more punctuation marks (>3) does not lead to the best performance of the model but does not have a significant impact on the original semantics of the utterance, and the model still outperforms the original SimCSE method.

As can be seen from Table 10, when only one symbol is inserted, the effect on the "roberta-base" of the model is negligible; when more symbols are inserted, the model is susceptible to poor robustness; when 1–4 symbols are inserted, the effect is weaker than the original model; when more symbols are inserted (6 or 7), the model tends to perform the same as the original model; when 1–2 symbols are inserted, the model has the best performance.

**Table 10.** Effect of the number of insertion symbols on the roberta-base model.

| Number of Symbols | ATEC | BQ | LCQMC | PAWS-X | STS-B | Mean |
|---|---|---|---|---|---|---|
| 0 | 34.11 | 50.18 | 65.39 | 10.27 | 29.54 | 37.89 |
| 1~1 | 32.78 | 50.83 | 65.83 | 10.04 | 29.78 | 37.78 |
| 1~2 | 33.90 | 51.52 | 67.76 | 10.92 | 29.56 | 38.73 |
| 1~3 | 32.89 | 50.41 | 66.36 | 10.45 | 30.00 | 38.02 |
| 1~4 | 31.20 | 50.79 | 66.55 | 9.83 | 29.91 | 37.65 |
| 1~5 | 31.74 | 50.54 | 67.92 | 10.42 | 30.11 | 38.14 |
| 1~6 | 32.40 | 50.58 | 66.26 | 10.61 | 29.73 | 37.91 |
| 1~7 | 33.68 | 50.85 | 66.02 | 9.20 | 30.19 | 37.98 |

As can be seen from Table 11, for the 'roberta-large' model, the performance of the model increases as the maximum number of symbols inserted increases from 1 to 5, with the best performance achieved when 1 to 5 symbols are inserted. There is no negative impact on the model when more symbols are inserted, and the model still outperforms the baseline (SimCSE).

**Table 11.** Effect of the number of insertion symbols on the roberta-large model.

| Number of Symbols | ATEC | BQ | LCQMC | PAWS-X | STS-B | Mean |
|---|---|---|---|---|---|---|
| 0 | 31.42 | 49.04 | 68.15 | 11.67 | 29.75 | 38.00 |
| 1~1 | 31.88 | 49.03 | 68.32 | 11.11 | 29.70 | 38.00 |
| 1~2 | 32.78 | 49.98 | 68.82 | 10.94 | 29.86 | 38.47 |
| 1~3 | 33.03 | 50.17 | 68.16 | 10.88 | 30.01 | 38.45 |
| 1~4 | 33.09 | 50.37 | 68.39 | 11.48 | 30.82 | 38.83 |
| 1~5 | 33.42 | 51.14 | 69.15 | 11.77 | 29.95 | 39.08 |
| 1~6 | 32.33 | 50.12 | 68.64 | 11.05 | 30.74 | 38.57 |
| 1~7 | 32.26 | 49.88 | 68.07 | 10.94 | 29.99 | 38.22 |

In summary, the bert-base model achieves the best performance when 1–3 punctuation symbols are inserted; the roberta-large model achieves the best performance when

1–5 symbols are inserted, and the insertion of more symbols does not negatively affect both models, and the model performance is still better than the baseline (SimCSE). The roberta-base model has the best performance when 1–2 symbols are inserted, and the model performance when more symbols are inserted is comparable to the baseline (SimCSE) effect.

*5.3. Chinese and English Punctuation and Choice of Symbol Types*

This section explored whether replacing English symbols with Chinese symbols would have an effect on the model. Furthermore, as question marks and exclamation marks may affect the tone or emotion of the sentence, the experiment also tried to remove these symbols. Then the effect of expanding the choice of more symbols (e.g., adding "[", "]", "~", "|", etc.) on the model. For the number of symbols inserted into the input text, the optimal number of symbols to be inserted for each model as experimented with in 5.2 was used.

In Table 12, for the "bert-base" model, the effect is weaker than that of English punctuation when the symbols are replaced with Chinese punctuation, indicating that Chinese punctuation has a greater influence on the semantics of sentences than English punctuation. However, if only Chinese punctuation is used, it still has a positive effect on the model. In addition, English exclamation marks and English question marks, which have a strong tone, do not have a significant impact on the semantics of Chinese sentences, and the addition of other special punctuation marks does not effectively improve the model performance.

**Table 12.** An exploration of Chinese and English punctuation-related issues based on the bert-base model.

| Operation | ATEC | BQ | LCQMC | PAWS-X | STS-B | Mean |
|---|---|---|---|---|---|---|
| no-operation | 33.38 | 50.51 | 62.00 | 9.46 | 27.29 | 36.52 |
| EdaCSE | 34.23 | 50.67 | 64.9 | 10.15 | 27.76 | 37.54 |
| Change it to Chinese symbols | 33.07 | 51.02 | 63.86 | 8.63 | 27.48 | 36.81 |
| The Chinese symbol is removed from the "!", "?" | 33.28 | 50.74 | 63.91 | 9.45 | 26.53 | 36.78 |
| The English symbol to remove the "!", "?" | 32.93 | 50.29 | 63.04 | 9.3 | 27.37 | 36.58 |
| Add additional English symbols | 33.1 | 51.15 | 62.53 | 9.49 | 26.37 | 36.52 |

In Table 13, for the "roberta-base" model, the performance of the model is weaker than that of the English punctuation model when the symbols inserted in the input of the model become Chinese symbols. The insertion of both Chinese punctuation and the strong semantic symbols in Chinese punctuation seems to have a negative impact on the semantic understanding of the sentence, making the insertion weaker than the original model. In contrast, strong punctuation in English does not negatively affect the model, but the positive effect on the model does not continue to grow when the choice of English symbols inserted is expanded.

**Table 13.** An exploration of Chinese and English punctuation-related issues based on the roberta-base model.

| Operation | ATEC | BQ | LCQMC | PAWS-X | STS-B | Mean |
|---|---|---|---|---|---|---|
| no-operation | 34.11 | 50.18 | 65.39 | 10.27 | 29.54 | 37.89 |
| EdaCSE | 33.90 | 51.52 | 67.76 | 10.92 | 29.56 | 38.73 |
| Change it to Chinese symbols | 32.74 | 51.42 | 66.72 | 8.63 | 29.48 | 37.79 |
| The Chinese symbol is removed from the "!", "?" | 31.49 | 51 | 67.23 | 9.98 | 29.33 | 37.80 |
| The English symbol to remove the "!", "?" | 33.93 | 51.29 | 65.04 | 10.3 | 29.37 | 37.98 |
| Add additional English symbols | 33.23 | 51.15 | 66.53 | 10.49 | 28.99 | 38.07 |

As shown in Table 14, the effect of adding Chinese punctuation to the input on the performance of the 'roberta-large' model is almost negligible, but it also indicates that the addition of Chinese punctuation does not have a beneficial effect on the model, and the strong tone symbols in Chinese do not seem to have much effect on the sentence meaning. Similar to the experimental results in the previous two models, the inclusion of strong tone

symbols in English punctuation has a positive effect on the model without affecting the semantics of the sentences, while the choice of too many English symbols still does not provide any additional benefit to the model.

**Table 14.** An exploration of Chinese and English punctuation-related issues based on the roberta-large model.

| Operation | ATEC | BQ | LCQMC | PAWS-X | STS-B | Mean |
|---|---|---|---|---|---|---|
| no-operation | 31.42 | 49.04 | 68.15 | 11.67 | 29.75 | 38.00 |
| EdaCSE | 33.42 | 51.14 | 69.15 | 11.77 | 29.95 | 39.08 |
| Change it to Chinese symbols | 31.74 | 51.42 | 66.72 | 10.63 | 29.48 | 37.99 |
| The Chinese symbol is removed from the "!", "?" | 31.49 | 51 | 67.23 | 10.98 | 29.33 | 38.00 |
| The English symbol to remove the "!", "?" | 33.93 | 51.29 | 66.04 | 11.3 | 29.37 | 38.38 |
| Add additional English symbols | 33.23 | 50.15 | 68.53 | 11.49 | 29.99 | 38.67 |

In summary, the introduction of Chinese punctuation does not improve the model performance as much as English punctuation and may even cause trouble for the model to understand the semantics of sentences, such as in the roberta-base model, while English punctuation such as exclamation marks and question marks, which have a strong tone of voice, does not have a negative impact on the model.

*5.4. Optimal Hyperparameter Experiments*

Experiments are carried out here on the values of the hyperparameter $\lambda$ in Equation (2), i.e., to explore the optimal ratio that each model needs to introduce for the information generated by the EdaCSE algorithm.

Table 15 shows that for the "bert-base" model, when a small proportion (less than 0.4) of the EdaCSE method is introduced, it does not have a positive impact on the model, while when the proportion is greater than or equal to 0.4, the performance of the model is significantly improved, and 0.6 is the optimal introduction proportion. For the "roberta-base" model, the best results were obtained by introducing the EdaCSE method at a scale of only 0.3, and for any value of the introduced scale, the model's results were improved compared with the original model (37.89). For the "roberta-large" model, it can be seen that the best results are obtained when we introduce the EdaCSE method at a large scale (0.6) to the model. As in the case of the "roberta-base" model, the introduction of the method at any scale has a positive effect on the original model to varying degrees.

**Table 15.** Selection of the value of the hyperparameter $\lambda$.

| Model | 0.1 | 0.2 | 0.3 | 0.4 | 0.6 | 0.8 | 1.0 |
|---|---|---|---|---|---|---|---|
| bert-base | 36.30 | 36.39 | 36.48 | 36.71 | 37.54 | 36.79 | 36.60 |
| roberta-base | 38.23 | 38.38 | 38.73 | 38.42 | 38.14 | 38.39 | 38.21 |
| roberta-large | 38.36 | 38.29 | 38.48 | 38.59 | 39.08 | 38.52 | 38.20 |

## 6. Conclusions

In this paper, we address the problem that the positive sample construction process in SimCSE may introduce a length bias to the model and propose an Easy Data Augmentation-Based Contrastive Learning of Sentence Embeddings (EdaCSE), which applies a perturbation to the length of the sentence by adding English punctuation to the original sentence, so that the model is no longer inclined to recognise sentences of similar length as similar sentences. Experiments demonstrate that the average performance of the three models on the five datasets is improved by 1.67, 0.84, and 1.08, respectively, after using the EdaCSE method compared with the original model.

The experiments also explored the number of punctuation symbols inserted and found that the bert-base model can achieve the best performance when inserting 1–3 punctuation symbols, the roberta-large model achieves the best performance when inserting 1–5 symbols, and the roberta-base model can have the best performance when inserting

1–2 symbols, with varying degrees of improvement in performance over the baseline model (SimCSE). In addition, through the experimental analysis of Chinese and English punctuation and symbol selection, it was found that the introduction of English punctuation can improve the performance of the bert-base, roberta-base, and roberta-large models, and the symbols with a strong tone, such as the English exclamation mark and the English question mark, do not have a great impact on the semantics of the Chinese sentences, whereas the model's performance is weaker with the introduction of Chinese punctuation than that of the introduction of English punctuation. Finally, based on the optimal hyperparameter experiments, it was found that introducing the information generated by the EdaCSE algorithm in any proportion will positively affect the original model to varying degrees and enhance the model's performance.

Therefore, the EdaCSE algorithm proposed in this paper can effectively improve the quality of the model-generated sentence representations, which provides a reference for the research of sentence representation algorithms.

**Author Contributions:** Conceptualization, W.G.; Methodology, X.L. (Xiaodong Liu); Writing—original draft, Y.L. (Yuxin Li) and Y.L. (Yanchi Li); Writing—review & editing, X.L. (Xiang Li). All authors have read and agreed to the published version of the manuscript.

**Funding:** This research was funded by research on adaptive integrated evolutionary algorithm for multi-root solution of complex nonlinear equations, grant number 62076225.

**Institutional Review Board Statement:** Not applicable.

**Informed Consent Statement:** Not applicable.

**Data Availability Statement:** Data sharing not applicable.

**Conflicts of Interest:** The authors declare no conflict of interest.

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
