# Peer review of "A Study of Contrastive Learning Algorithms for Sentence Representation Based on Simple Data Augmentation"

_applsci, doi:10.3390/app131810120_

Round 1

Reviewer 1 Report

This paper presents a simple data augmentation for learning sentence representation. The proposed method called EdaCSE, perturbs the input sentence by adding punctuation marks. As a result, the sentence length changes. 

I have the following concerns:

1. As the main idea of this paper is by adding punctuation marks, Section 3.2 only gives a few examples in Figure 1 to see the perturbed sentences. However, details of adding punctuation marks are not given. For example, how to decide the positions for adding punctuation marks, and what punctuation marks are inserted?

2. In Eq. (1), the loss is contrastive learning loss which consists of losses of SimCSE and EdaCSE. It would be helpful to give the overall loss when learning sentence representation.

3. In Table 1, as +EdaCSE represents the result of using both losses of SimCSE and EdaCSE, it is not clear the contribution of EdaCSE loss. 

4. In the abstract, it says "meaningless English punctuation marks". First, the punctuation marks are  not meaningless. Second, in the experimentation and example in Figure 1, Chinese punctuation marks are used for Chinese sentence representation learning. 

Reviewer 2 Report

The paper address the SimCSE algorithm problem where the greater the difference in sentence length, the smaller the probability that the sentence pairs are similar. An EdaCSE algorithm is proposed to perturb the sentence length using a simple data enhancement method without affecting the semantics of the sentences.

It will help the reader to get a bird-eye view of the related work with a table listing the previous work.

The dataset should also be listed in a table.

The evaluation metrics should be explained and clearly stated in Table 1.

The structure of the paper should be organized well. Why the authors discuss the result in Table 1 even before section 4?

As shown in Table 1, why the model performance for PAWS-X dataset is worse than the performance for STS-B dataset even though STS-B dataset is the smallest. An error analysis should be conducted on the PAWS-X dataset before continuing the experiments. Table 3, 4, and 5 show that the model has good performance on the size of 10,000, even though the total size of combined dataset is 438,650. An error analysis should be conducted or a new good dataset should be utilized.

The paper is trying to overcome the problem of SimCSE algorithm: "a bias into the model that incorrectly assumes that the more similarly lengthed samples are more likely to be positive samples, and the greater the difference in length the smaller the probability that the sentence pairs are similar. The authors propose to add punctuation marks without changing the meaning of the original text. However, Table 7-9 show that inserting more than 5 symbols will negatively affect the performance. Again, an error analysis should be conducted. The authors should also explain why focus on adding certain number of symbols rather than focus on adding symbols and checking the difference between the length of the original and augmented text? 

Overall, the paper seems like a series of experiment conducted by the authors without a proper research design. Hence, the conclusion is also inconclusive. 

Reviewer 3 Report

The authors described an attempt to address a limitation found in SimCSE algorithm. Although some readers of Appl. Sci. may find this interesting, the contribution of this paper is not clear and the article is very difficult to understand due to poor grammar. The authors simply considered a single limitation of SimCSE (based on vector length) and presented a comparative study result with their approach which added noise in form of punctuations. This is not sufficiently evaluated to show that results of comparison was indeed due to the perturbation. The introduction and literature are very shallow and does not identify specific challenges addressed by the research. Numerous claims were made throughout the article without supporting evidence. Overall, the paper lacks novelty and scientific rigour required for journal publication.

The English is very poor and requires substantial editing

Round 2

Reviewer 1 Report

I would like to thank the authors for considering my comments. Most of my concerns in the previous review are well addressed.  I have two further suggestions for the presentation. 

1) To make the description in lines 179~184 simple, I suggest naming rows in Table 1 as #1 ~ #6.
2) In Eq. (2), it needs to give the formal definition of the two contrastive losses, simply as the Eq. 1 in https://arxiv.org/pdf/2104.08821.pdf. 

Reviewer 2 Report

The authors have addressed my comments

Author Response

Thank you for your comments and suggestions !

Reviewer 3 Report

The authors made no attempt to address the concerns raised. There is limited evidence to justify that results of comparison was indeed due to the perturbation. 

The introduction and literature were very shallow and does not identify specific challenges addressed by the research. This is still the same.

Numerous claims were made throughout the article without supporting evidence. Overall, the paper lacks novelty and scientific rigour required for journal publication.

The authors provided a one line response below.

Response 1: Thank you for the review comments! We will revise and improve these issues.

The English is very poor 

Author Response

Thank you for your comments and suggestions, we will endeavour to amend the issues you have raised.